# A “Pincer” Type of Acridine–Triazole Fluorescent Dye for Iodine Detection by Both ‘Naked-Eye’ Colorimetric and Fluorometric Modes

**DOI:** 10.3390/molecules29061355

**Published:** 2024-03-19

**Authors:** Mei Yu, Lu Jiang, Lan Mou, Xi Zeng, Ruixiao Wang, Tao Peng, Fuyong Wu, Tianzhu Shi

**Affiliations:** 1Department of Brewing Engineering, Moutai Institute, Renhuai 564500, China; yumei@mtxy.edu.cn (M.Y.); jianglu@mtxy.edu.com (L.J.); moulan@mtxy.edu.cn (L.M.); zengxi@mtxy.edu.com (X.Z.); wangruixiao@mtxy.edu.com (R.W.); edifcztony@126.com (T.P.); 2Key Laboratory of Macrocycle and Supramolecular Chemistry of Guizhou Province, Guiyang 550025, China

**Keywords:** fluorescent probe, naked eye detection, iodide ions, tweezer-type acridine–triazole, dual-detection method

## Abstract

Iodine, primarily in the form of iodide (I^−^), is the bioavailable form for the thyroid in the human body. Both deficiency and excess intake of iodide can lead to serious health issues, such as thyroid disease. Selecting iodide ions among anions has been a significant challenge for decades due to interference from other anions. In this study, we designed and synthesized a new pincer-type acridine–triazole fluorescent probe (probe **1**) with an acridine ring as a spacer and a triazole as a linking arm attached to two naphthol groups. This probe can selectively recognize iodide ions in a mixed solvent of THF/H_2_O (*v*/*v*, 9/1), changing its color from colorless to light yellow, making it suitable for highly sensitive and selective colorimetric and fluorescent detection in water systems. We also synthesized another molecular tweezer-type acridine–triazole fluorescent probe (probe **2**) that exhibits uniform detection characteristics for iodide ions in the acetonitrile system. Interestingly, compared to probe **2**, probe **1** can be detected by the naked eye due to its circulation effect, providing a simple method for iodine detection. The detection limit of probe **1** is determined to be 10^−8^ mol·L^−1^ by spectrometric titration and isothermal titration calorimetry measurements. The binding stoichiometry between probe **1** and iodide ions is calculated to be 1:1 by these methods, and the binding constant is 2 × 10^5^ mol·L^−1^.

## 1. Introduction

Anions play crucial roles in life health, pharmacy, molecular catalysis, and environmental sciences [1,2,3,4]. Abnormal levels of small molecules, such as iodide ions (I^−^), in biological systems could be critical indicators of certain diseases and pose risks to human physiological functions [5,6,7]. Therefore, the sensitive detection of iodine ions has become increasingly important in these fields [8]. Traditional techniques have been employed for iodine detection [9,10,11,12], but their application is limited due to the need for skilled technicians and high costs. Consequently, there is a pressing need to design and synthesize highly specific sensors with superior performance to meet practical application requirements. Colorimetric fluorescence, a novel detection method, is garnering significant attention from researchers [13,14,15,16]. Several sensitive fluorescent probes for iodide detection, based on changes in fluorescence intensity and colorimetry, have been reported [17,18,19,20,21]. These studies provide valuable insights for future research [22]. However, designing and synthesizing a colorimetric probe with a simple structure, excellent sensitivity, and selectivity for the visual detection of trace I^−^ in complex systems remains a challenge [23,24,25,26].

The molecular pincer probe, characterized by its two arms connected by a rigid spacer group, exhibits both flexibility and rigidity. The terminal arm can be modified with various functional groups, rendering this type of molecule easily adaptable and synthetically versatile [27,28,29]. These fluorophores not only provide multiple coordination points but also enhance the adjustability of the clamp-type space. For instance, 4,5-bis[(*N*,*N*-dihydroxyethyl)aminomethyl)] acridine has been reported as a fluorescent reagent with a fluorescent switch type for the detection of Cd^2+^-Cu^2+^/I^−^-Hg^2+^. Acridine, when attached with azacrown ether and azasulfide, can detect Hg^2+^ and Cd^2+^. Moreover, 4,5-bis(*N*,*N*-bis(pyridine-2-methyl) Aminomethyl Acridine can detect PF_6_^−^ and also recognize CN^−^ and H_2_PO_4_^−^, among others [30,31,32,33,34]. Research has shown that the planar rigid skeleton structure and the easily modifiable chemical properties of the acridine ring make it an ideal molecular tweezer spacer for the design and synthesis of different functional chromophores.

Inspired by the unique molecular space structure’s detection performance, this article presents two different molecular tweezers probe compounds, **1** and **2**, with acridine and their recognition performance for the I^−^ ion. By leveraging the tunability of UV-vis and fluorescence spectroscopy, probes **1** and **2** were used to determine the recognition performance and the mixing ratio with iodide ions. The mechanism study revealed that the probe reacted with iodine by forming a hydrogen bond between I^−^ and the aromatic hydrogen on the triazole ring in probe **1**, enabling colorimetric detection. The probe can also be reused with silver ions in the competitive reaction.

## 2. Results

### 2.1. Spectra Property

Upon the addition of various anions, no significant changes were observed in the UV-vis spectra, except for I^−^ at 270 nm~500 nm, as depicted in Figure 1a. Interestingly, the interaction of **1** with I^−^ resulted in the emergence of two new absorbance peaks at 295 nm, while the absorbance peaks at 265 and 363 nm significantly increased. This differential response of **1** to anions could be utilized to develop a colorimetric assay. The color transitioned from colorless to light yellow within 10 s, a change sufficiently distinct to be discernible by the naked eye. Similarly, upon the addition of anions to probe **2** (10 µmol·L^−1^, acetonitrile), only the addition of I^−^ significantly enhanced the probe’s absorption at 254 nm, accompanied by a blue shift of 7 nm (Figure 1b). The crystal data of Probe **2** reveals that the two hydrogens on the triazole are on the same side, not the inner side, which reduces the system energy (Appendix A). Interestingly, both probe **1** and probe **2** exhibited maximum absorption bands at approximately 250 nm, which slightly blue-shifted following the addition of the I^−^ ion. This can be attributed to the increase in the system’s energy induced by the I^−^ ion. Compared to the electron-withdrawing pyridine configuration of probe **2**, the electron-donating naphthyloxy group of probe **1** exhibits a stronger intramolecular charge transfer (ICT) effect. Concurrently, the coplanarity of the molecule is increased by the I^−^ ion induction. These factors contribute to the marked difference in the absorption of probe **1** at 300–400 nm in the UV absorption (Figure 1a,b).

The PL of probe **1** remained unchanged with the addition of other anions, maintaining a maximum emission band at 427 nm upon excitation at 357 nm. The quantum ratio in THF/H_2_O (*v*/*v*, 9/1) was determined at 25 °C. Upon the addition of I^−^ to the solution, a fluorescence quenching of 77% was observed, and the emission peak underwent a minor red shift from 427 nm to 437 nm. Subsequently, a new emission peak emerged, suggesting that the new species is less luminescent than the original one. However, the addition of other anions resulted in no significant changes in the fluorescent spectra (Figure 2a). A similar phenomenon was observed in the acetonitrile solution of probe **2** (10 µmol·L^−1^), with 247 nm as the excitation wavelength and 425 nm as the emission wavelength. Besides the weak reduction of fluorescence by F^−^ and AcO^−^, the addition of iodide ions notably quenched the fluorescence of the probe (the quenching rate is 70%, Figure 2b).

The UV-vis and fluorescence emission spectra of probe **1** (5.0 × 10^−5^ mol·L^−1^, THF/H_2_O, *v*/*v*, 9/1) and probe **2** (5.0 × 10^−5^ mol·L^−1^, acetonitrile) were recorded after mixing different concentrations of spectral titration, a series of probes, and adding I^−^ solutions of different concentrations. The probe concentration was kept at 1.0 × 10^−5^ mol·L^−1^ in the molar ratio method. In the Job method, the total concentration of the probe and anion was kept constant at 8 × 10^−5^ mol·L^−1^ during absorption spectrometry and 5 × 10^−5^ mol·L^−1^ during fluorescence spectrometry.

A titration of **1** with I^−^ indicated a gradual increase in the absorbance at 300 nm and 357 nm upon the addition of I^−^ concentration up to 30 equivalents, as shown in Figure 3a. The increase in I^−^ concentration led to two different species bonding probabilities, with two clear absorbance points at 300 nm and 357 nm observed during the titration process, indicating a weak interaction of I^−^ with receptor **1**.

The stoichiometry between **1** and I^−^ was determined using the molar ratio method and Job’s plot analyses. This involved maintaining the total concentration of **1** and I^−^ at 30.0 μM and varying the molar ratio of I^−^ (XM; XM = [I^−^]/{[1] + [I^−^]}) from 0 to 1.0. At a molar fraction of I^−^ of 0.5, the absorbance at 357 nm reached its maximum, indicating the formation of a 1:1 complex between **1** and I^−^ (Figure 3 inset). The binding stoichiometry and the binding constant to I^−^ were determined using the Benesi–Hildebrand double reciprocal method, as per Equation (1) [35,36]. For probe **1**, a linear fit was obtained using Equation (1), indicating a binding stoichiometry of 1:1, consistent with the presence of two triazole groups in probe **1**. The binding constant was found to be 2 × 10^5^ L·mol^−1^. Both the spectral titration molar ratio method and the Job method confirmed that the ratio of probe **1** to I^−^ is 1:1 (Figure 3a,b inset). This observation was attributed to the heavy atom effect of I^−^ and the modulation by the extension of π-conjugation as a result of iodine binding through the triazole of receptor **1** [37]. Figure 3b shows the titration photographs of 1 in the presence of different I^−^ concentrations. As the I^−^ concentrations increased, the fluorescence of the probe reached 86% after adding 40 times the amount of different anions relative to the probe. This was in agreement with the rising absorption band at approximately 300 nm and 357 nm in the corresponding UV/Vis spectra, demonstrating the feasibility of naked-eye detection (Figure 3c inset). Thus, probe **1** was identified as a selective and sensitive colorimetric probe for I^−^. In comparison, probe **2** underwent the same test in a CH_3_CN solution, further validating our hypothetical mechanism. The results reveal a fluorescence quenching probe with I^−^ quencher based on the static quenching mechanism.

To investigate the selectivity of probe **1** for I^−^ in the presence of other anions, competitive anion titrations were conducted. Probe **1** was exposed to 20.0 equivalents of I^−^ alongside 20.0 equivalents of various other anions. The results demonstrated that the presence of different anions did not significantly alter the spectral characteristics, and the fluorescence intensity remained consistent with I^−^ ions, as depicted in Figure 4a. Consequently, the selectivity of probe **1** for I^−^ was not affected by the presence of other anions. These findings confirm that probe **1** can serve as a novel chemosensor for I^−^, offering both colorimetric and fluorescence detection capabilities. A similar outcome was observed for the probe **2**-I^−^ complex (represented by the gray bars in Figure 4b). Both probes **1** and **2** exhibited excellent selectivity for I^−^ ions.

### 2.2. Reversibility and Reusability of Probe ***1***

Most I^−^ ion chemosensors available in the literature are irreversible and have limited usage [38,39,40]. The development of a reversible and reusable sensor for selective anion detection would be highly beneficial. To evaluate the reversibility and reusability of probe **1**, we conducted systematic titration studies of the **1**-I^−^ complex, gradually adding increasing amounts of Ag^+^ (silver perchlorate). These studies were performed using fluorescence techniques in a mixed solution of THF/H_2_O (*v*/*v*, 9/1). The fluorescence spectral titration of the **1**-I^−^ complex with incremental additions of Ag^+^ (0–80 equiv.) in the THF/H_2_O solution is depicted in Figure 5. As Ag^+^ was added to the **1**-I^−^ complex, the fluorescence emission at 427 nm gradually increased. This suggests that the addition of Ag^+^ displaced I^−^ from the complex, resulting in the formation of AgI. Upon the addition of I^−^ ions, the heavy atom effect and the electron-rich ethylene moiety, due to deprotonation, engaged in photoinduced electron transfer, quenching the excited state of the triazole. However, upon titration with Ag^+^ this quenching was prevented, and the fluorescence properties were restored (see inset in Figure 5). The titration of probe **1** with I^−^ ions resulted in fluorescence quenching due to the formation of the **1**-I^−^ complex, acting as an OFF switch. Conversely, titration of the **1**-I^−^ complex with Ag^+^ led to the protonation of the triazole and the reformation of probe **1**. This was accompanied by a significant increase in fluorescence intensity, acting as an ON switch (Figure 5).

### 2.3. Mechanism Tests

#### 2.3.1. Isothermal Calorimetric Titration (ITC) Studies

ITC is a conventional characterization method that uniquely measures binding reactions by detecting heat changes during the formation of complexes in various physicochemical processes. While ITC has been extensively reported in research involving metal ions and biomolecules [41,42,43,44], there are fewer reports concerning anions. The ITC data analysis software (v1.41) employs a non-linear regression, an iterative process where model variables are determined by making a series of approximations to the ITC titration curve. This process identifies the system’s best-fit parameters (H, Ka, and N) [45]. Recently, we used ITC to measure the binding thermodynamics and evaluate the binding mechanism between the probe and I^−^ (Figure 6). The independent binding site model was used to fit the action constant of probe **1** and the I^−^ binding process, with Ks = (2.000 ± 0.92) × 10^5^ L·mol^−1^, the number of action sites *n* = (0.976 ± 0.063) (a value close to the binding stoichiometry of 1:1), the molar binding enthalpy Δ*H* = (126.3 ± 15.54) KJ·mol^−1^, and molar binding entropy ΔS = 525.0 J·mol^−1^·K^−1^. The calculated molar binding free energy Δ*G*^Θ^ = −(156.40 ± 15.54) kJ·mol^−1^ indicates that the coordination reaction is exothermic and spontaneous. All these results suggest that the binding of probe **1** and I^−^ is spontaneous and thermodynamically feasible. However, as shown in Figure 6a, as the concentration of the probe and iodine binding increases, the system gradually releases heat, which may be attributed to their relatively weak binding force. The trend in the ITC graph also reflects consistency with spectral data. All of the ITC experiments were endothermic (ΔH > 0), and the stoichiometry (values *n*) was approximately **1**. The evaluated binding constants for probe **1** with I^−^ are in good agreement with those obtained by fluorescence titration under the same conditions.

#### 2.3.2. ^1^H NMR Study

In the field of anion recognition mechanisms, the use of ^1^HNMR resonance spectroscopy to investigate hydrogen bonding within the recognition system is a well-established method. It provides direct evidence of the role of anion complexes. The interaction between probe **1** and iodide (I^−^) was examined using 1HNMR titration, as depicted in Figure 7A. As the concentration of I^−^ in probe **1** increased, the absorption peaks at chemical shifts 7.931 and 7.512 moved to the low field. The Hb proton peak at 7.512 is characteristic of the triazole ring’s aromatic hydrogen. Upon the addition of increasing quantities of I^−^ to sensor **1** (up to 3.0 equiv.), the aromatic hydrogen proton shifted 0.016 ppm to 7.528, and Ha shifted 0.009 ppm to 7.931 ppm. These shifts can be attributed to the C–H···I hydrogen bonding, which enhances the electron-donating ability of the ethylene moiety through deprotonation. This, in turn, facilitates photoinduced electron transfer to the excited triazole core, resulting in a shift of the chemical value to a low field and a fluorescence turn-off. The heavy atom effect was also taken into consideration. Similarly, the same interaction and hydrogen bond interaction between the triazole aromatic hydrogen and the anion were observed in probe **2** (Appendix A).

### 2.4. Practical Application

Given the complexity of real-world factors that can affect the quality and stability of probes, the detection of iodide ions (I^−^) is both important and challenging. Therefore, it is desirable to have a straightforward method for detecting I^−^ ions in various types of water. Experiments were conducted using probes **1** and **2** to potentially detect I^−^ ions in deionized water, lake water, pond water, and tap water. The standard curve method was employed to test the recovery rate of I^−^ ions in each water sample using fluorescence analysis. The results of these analyses are presented in Table 1. The experimental findings indicate that the recovery rate ranges between 90% and 96% in Table 2. These fluorescence studies suggest that probe **1** offers certain advantages at low concentrations for the detection of I^−^ ions in different water contents within raw sample materials.

## 3. Materials and Methods

### 3.1. Chemicals and Instruments

Cary Eclipse Fluorescence Spectrophotometer (Varian, Palo Alto, CA, USA); TU-1901 UV-Vis Spectrophotometer (Beijing Puxi General Instrument Co., Ltd., Beijing, China); Nova-400 Nuclear Magnetic Resonance Spectrometer (Varian); LC/MSD Mass Spectrometer (Agilent, Santa Clara, CA, USA); Vertex 70 FTIR infrared spectrometer (Bruker, Billerica, MA, USA); Nano isothermal calorimetric titrator (TA instrument, New Castle, DE, USA); X-5 digital display microscopic melting point tester (Beijing Tektronix Instruments Co., Ltd., Beijing, China, temperature uncorrected).

The anion solution was synthesized using tetrabutylammonium salt (Shanghai Jingchun Reagent Co., Ltd., Shanghai, China) in a solvent mixture of tetrahydrofuran and acetonitrile. The reagents used in the experiment were of analytical grade, and the water was doubly distilled.

### 3.2. Design and Synthesis

A convenient synthetic route to chemosensor **1** was developed, as shown in Figure 1. The raw material, acridine, is used to prepare 4,5-bis(azidomethyl)acridine through a two-step reaction. This compound then reacts with the intermediates 3-(2-naphthyloxy)propyne and 2-pyridineacetylene. The resulting acridine fluorescent probes **1** and **2** feature an acridine ring as the spacer and a triazole as the linking arm, symmetrically connecting two naphthol groups and pyridine groups. The synthesis process is as follows: Under a nitrogen atmosphere, 5.58 mmol of acridine and 25 mL of concentrated sulfuric acid are added to a 100 mL three-necked flask. The mixture is heated at 50 °C until completely dissolved, then 22.32 mmol (1.8 g) of chloromethyl methyl ether is added dropwise, and the reaction is allowed to proceed for 36 h. After cooling to room temperature, the mixture is poured into 200 mL of ice water for suction filtration, extracted with chloroform, washed three times with 30 mL of water, dried with anhydrous magnesium sulfate, and filtered, and the solvent is evaporated. The resulting product, 4,5-bis(chloromethyl)acridine, is purified by column chromatography on silica.

In a 100 mL three-necked flask, 0.725 mmol of the compound 4,5-bis(chloromethyl)acridine, 0.2 g of sodium azide, and 40 mL of *N*,*N*-dimethylformimide are combined and reacted at 70 °C for 12 h. After cooling to room temperature, the mixture is poured into 200 mL of ice water and left to stand overnight. The precipitate is collected by suction filtration and dried in a vacuum to yield 0.23 g of 4,5-bis(azidomethyl)acridine. This crude product can be used in the next reaction. In another 100 mL three-necked flask, 50 mL of 2-naphthol (4.4 mmol) and anhydrous potassium carbonate (4 mmol) are heated to reflux for 0.5 h in acetone. After cooling, bromopropyne (4 mmol) is added dropwise, and the mixture is refluxed for 24 h. After cooling to room temperature, the mixture is filtered and the solvent is removed. The product, 3-(2-naphthyloxy)propyne, is obtained by column chromatography on silica using pure chloroform as the eluent. This product is used in the next step of the synthesis, as shown in Figure 1 (Appendix A).

In a 100 mL three-necked flask, 4,5-bis(azidomethyl)acridine (0.35 mmol) was combined with CuI (0.1 mmol) and 3-(2-naphthyloxy)propyne (0.88 mmol). This mixture was dissolved in a 40 mL solution of THF/H_2_O (3/1, *v*/*v*) and heated to reflux for 12 h under a nitrogen atmosphere. The solution was then concentrated to remove the THF, followed by extraction with chloroform (3 × 20 mL). The extract was washed with saturated brine (20 mL), dried over anhydrous MgSO_4_, and the solvent removed. The residue was purified by column chromatography on silica to yield a pale yellow solid (170 mg, 74% yield). m.p. 218–219 °C; ^1^H NMR (CDCl_3_, 400 MHz) δ (ppm): 9.23 (s, 1H, Ar-H), 8.39 (s, 2H, Ar-H), 8.22 (d, 2H, J = 9.0 Hz, Ar-H), 7.79~7.61 (m, 10H, Ar-H), 7.41~7.37 (m, 4H, Ar-H), 7.34~7.29 (m, 2H, Ar-H), 7.11 (dd, 2H, J = 2.8 Hz, 2H), 6.34 (s, 4H, O-CH_2_-), 5.21 (s, 4H, -CH_2_-); IR (KBr, cm^−1^): 2935, 1716, 1626, 1465, 1391, 1216, 1011, 838, 763; ESI-MS: *m*/*z* 654 [M + H]^+^.

Following the above experimental procedure, we obtained probe **2** with a yield of 54% (93 mg). m.p. 214~217 °C; ^1^H NMR (CDCl_3_, 500 MHz) δ(ppm): 8.84 (s, 1H, Ar-H), 8.45 (d, 2H, *J* = 4 Hz, Ar-H), 8.22 (s, 2H, Ar-H), 8.14 (d, 2H, *J* = 7.5 Hz, Ar-H), 8.04 (d, 2H, *J* = 1 Hz, Ar-H), 7.75 (d, 2H, *J* = 4.5 Hz, Ar-H), 7.71 (t, 2H, *J* = 1.5 Hz, Ar-H), 7.56~7.53 (m, 2H, Ar-H), 7.15 (m, 2H, Ar-H), 6.41 (s, 4H, Ar-CH_2_-); ^13^C NMR (CDCl_3_, TMS, ppm): 150.4, 149.2, 148.3, 146.1, 137.1, 136.2, 133.1, 131.1, 129.4, 126.7, 125.8, 122.6, 122.6, 120.2, 50.7; IR (KBr, cm^−1^): 3053, 1720, 1600, 1537, 1465, 1421, 1356, 1225, 1149, 1081, 1045, 993, 916, 832, 785, 752, 661, 611, 515; ESI-MS: *m*/*z*+: 496.2 [M + H]^+^.

### 3.3. Methodology

Spectral Properties: The UV-vis spectra were obtained from a 1 mL stock solution of probe **1** (1.0 × 10^−4^ mol·L^−1^). This solution was transferred into a 10.0 mL volumetric flask, to which 1 mL of water and 1 mL of an anion solution (2 × 10^−3^ mol·L^−1^) were added. The anions used were I^−^, F^−^, Cl^−^, Br^−^, NO_3_^−^, HSO_4_^−^, ClO_4_^−^, PF_6_^−^, AcO^−^, and H_2_PO_4_^−^. The solution was then diluted with THF (10 µmol·L^−1^, THF/H_2_O, *v*/*v*, 9/1), thoroughly mixed, and left at room temperature for 30 min. The solution was then transferred to a 1 cm quartz cuvette for UV-vis and fluorescence spectrometry. Fluorescence spectroscopy measured the excitation and emission wavelengths at 357/427 nm. A similar measurement system was used for the control of probe **2** (10 µmol·L^−1^, acetonitrile).

Isothermal Calorimetric Titration (ITC) Studies: Probe **1** was dissolved in THF at room temperature (298.15 K) to prepare a stock solution (10^−4^ M). Test solutions were prepared by adding 10 μL of the stock solution to a 250 μL syringe, followed by the addition of a uniform aliquot of the test solution to the isothermal titration microcalorimeter. After titration, the heat of reaction was corrected and graphed. The data was then fitted to obtain a curve representing the heat of the titration reaction. The ITC experiment condition is as follows: agitator speed, 300 rpm; number of titrations, 25; interval, 300 s; reaction temperature, 298.15 K.

## 4. Conclusions

The compound 4,5-bis(azidomethyl)acridine is synthesized from acridine and subsequently reacts with intermediates 3-(2-naphthyloxy) propyne and 2-pyridineacetylene. This reaction yields acridine fluorescent probes **1** and **2**, which possess a clamp-type structure that symmetrically connects two naphthol groups and pyridine groups. These probes can be utilized to detect trace iodide ions (I^−^) using ultraviolet-visible absorption spectroscopy and fluorescence spectroscopy. However, probe 1 exhibits several advantages:Unlike most I-probe complexes, the spectroscopic detections of probe **1** can be performed in a THF/H_2_O system, offering broader applicability compared to detection in a purely organic system.I^−^ can change the solution of probe **1** from colorless to light yellow, providing a visual detection capability.Despite being a fluorescence-quenching type, the detection limit of probe **1** still reaches 10^−8^ mol·L^−1^ The linear range of the I^−^ concentration detected by the two spectral techniques spans two orders of magnitude.In fluorescence detection, the addition of an appropriate amount of silver ions (Ag^+^) can compete to release probe **1**, demonstrating a reversible effect.

Thus, probe 1 exhibits the potential for reuse as a detection reagent and serves as an excellent tool for the fluorescence and colorimetric detection of water-soluble trace iodide ions.

## Data Availability

Data are contained within the article and Appendix A.

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
