# Peer review of "A “Pincer” Type of Acridine–Triazole Fluorescent Dye for Iodine Detection by Both ‘Naked-Eye’ Colorimetric and Fluorometric Modes"

_molecules, 2024, doi:10.3390/molecules29061355_

Round 1

Reviewer 1 Report

Comments and Suggestions for Authors

The idea that the authors present in the article is good. However, they lack the presentation of control results, and in the fluorescence section, they need to demonstrate that the observed fluorescence quenching is indeed due to PET (Photoinduced Electron Transfer) and not other phenomena like Dexter. They also did not characterize the phenomenon as static or dynamic. The article cannot be accepted without addressing these issues.

-In the abstract, the authors wrote, "The selection of iodide ions among anions has been a significant challenge for decades, because other anions often interfere with the detection process." The authors should provide a brief explanation of the type of detection and the importance of iodide ions in relation to other ions in the processes to provide context for the readers.

-When citing references, the authors consistently place a comma before the references. The references should be presented without a comma.

-In line 31, the authors wrote, "The sensitive detection of iodine ion becomes urgent and important in these fields,[5]." The authors should elaborate more on why the detection of iodine is so crucial.

-Lines 65 to 73 should be moved to the methodology section instead of the results.

-The graphs in Figures 1 and 2 are difficult to understand. Improve the quality of the graphs and add color to distinguish different spectra.

-The control spectrum of only probes 1 and 2 individually, without ions, is not clearly identified. The authors should highlight the control or add it to the figure. The control of the absorbance of individual ions should also be included.

-The authors showed fluorescence quenching but did not show the correction for internal filter effects. The authors should include this experiment in the manuscript.

-In lines 138-140, the authors mention PET modulation. How can the authors be sure it is PET and exclude other phenomena like Dexter effects? Was any time-resolved fluorescence performed to confirm which phenomenon is occurring?

-In the fluorescence experiments, the authors should characterize whether the phenomenon is static or dynamic.

-Improve the quality of Figure 3; the insets are barely visible. Add colors for better clarity.

-Lines 200 to 204 should be moved to the methodology section.

-The results of ITC in Figure 6 are described in the figure caption. The authors should include this information in the results section and discuss it.

-All information related to absorbance, fluorescence, and ITC experiments is missing. The authors did not describe how the experiments were conducted, how errors were generated, and whether there were duplicates or triplicates.

 -The authors should add conclusions to their work.

Comments on the Quality of English Language

Moderate editing of English language required

Author Response

Dear  Reviewer:

Thank you for your letter and for the reviewers’ comments concerning our manuscript entitled “A “Pincer” Type of Acridine-Triazole Fluorescent Dye for Iodine Detection by Both ‘Naked-eye’ Colorimetric and Fluorometric Modes” (Manuscript Number: molecules-2877702).

    Here we submit the revised version where all concerns from reviewers are fully addressed and revised (see the list of point-by-point response below; the corresponding response and revision in the manuscript have been highlighted in red color).

We are looking forward to your decision.

Best regards

Tianzhu  Shi 

Reviewer 2 Report

Comments and Suggestions for Authors

Report attached as a PDF file.

Comments on the Quality of English Language

The language should be improved.

Author Response

Thank you for your letter and for the reviewers’ comments concerning our manuscript entitled “A “Pincer” Type of Acridine-Triazole Fluorescent Dye for Iodine Detection by Both ‘Naked-eye’ Colorimetric and Fluorometric Modes” (Manuscript Number: molecules-2877702).

    Here we submit the revised version where all concerns from reviewers are fully addressed and revised (see the list of point-by-point response below; the corresponding response and revision in the manuscript have been highlighted in red color).

We are looking forward to your decision.

Best regards

Tianzhu  Shi

Round 2

Reviewer 1 Report

Comments and Suggestions for Authors

Yu, M. et al. made the modifications to the manuscript as requested. From my part, the article can be published.

Reviewer 2 Report

Comments and Suggestions for Authors

I have found the current version of the manuscript improved. Moreover, the questions from me to the authors are answered. Thus, I have no objections to accept the manuscript for publication.